# FreeLB: Enhanced Adversarial Training for Natural Language Understanding

**Chen Zhu[1], Yu Cheng[2], Zhe Gan[2], Siqi Sun[2], Tom Goldstein[1], Jingjing Liu[2]**
[1]University of Maryland, College Park    [2]Microsoft Dynamics 365 AI Research
{chenzhu,tomg}@cs.umd.edu, {yu.cheng,zhe.gan,siqi.sun,jingjl}@microsoft.com

## Abstract

Adversarial training, which minimizes the maximal risk for label-preserving input perturbations, has proved to be effective for improving the generalization of language models. In this work, we propose a novel adversarial training algorithm, FreeLB, that promotes higher invariance in the embedding space, by adding adversarial perturbations to word embeddings and minimizing the resultant adversarial risk inside different regions around input samples. To validate the effectiveness of the proposed approach, we apply it to Transformer-based models for natural language understanding and commonsense reasoning tasks. Experiments on the GLUE benchmark show that when applied only to the finetuning stage, it is able to improve the overall test scores of BERT-base model from 78.3 to 79.4, and RoBERTa-large model from 88.5 to 88.8. In addition, the proposed approach achieves state-of-the-art single-model test accuracies of 85.44% and 67.75% on ARC-Easy and ARC-Challenge. Experiments on CommonsenseQA benchmark further demonstrate that FreeLB can be generalized and boost the performance of RoBERTa-large model on other tasks as well. [1]

## 1    Introduction

*Adversarial training* is a method for creating robust neural networks. During adversarial training, mini-batches of training samples are contaminated with adversarial perturbations (alterations that are small and yet cause misclassification), and then used to update network parameters until the resulting model learns to resist such attacks. Adversarial training was originally proposed as a means to enhance the security of machine learning systems (Goodfellow et al., 2015), especially for safety-critical systems like self-driving cars (Xiao et al., 2018) and copyright detection (Saadatpanah et al., 2019).

In this paper, we turn our focus away from the security benefits of adversarial training, and instead study its effects on generalization. While adversarial training boosts the robustness, it is widely accepted by computer vision researchers that it is at odds with generalization, with classification accuracy on non-corrupted images dropping as much as $10\%$ on CIFAR-10, and $15\%$ on Imagenet (Madry et al., 2018; Xie et al., 2019). Surprisingly, people observe the opposite result for language models (Miyato et al., 2017; Cheng et al., 2019), showing that adversarial training can improve both generalization and robustness.

We will show that adversarial training significantly improves performance of state-of-the-art models for many language understanding tasks. In particular, we propose a novel adversarial training algorithm, called FreeLB (Free Large-Batch), which adds adversarial perturbations to word embeddings and minimizes the resultant adversarial loss around input samples. The method leverages recently proposed "free" training strategies (Shafahi et al., 2019; Zhang et al., 2019) to enrich the training data with diversified adversarial samples under different norm constraints at no extra cost than PGD-based (Projected Gradient Descent) adversarial training (Madry et al., 2018), which enables us to perform such diversified adversarial training on large-scale state-of-the-art models. We observe improved invariance in the embedding space for models trained with FreeLB, which is positively correlated with generalization.

---

[1]Code is available at https://github.com/zhuchen03/FreeLB.

We perform comprehensive experiments to evaluate the performance of a variety of adversarial training algorithms on state-of-the-art language understanding models and tasks. In the comparisons with standard PGD (Madry et al., 2018), FreeAT (Shafahi et al., 2019) and YOPO (Zhang et al., 2019), FreeLB stands out to be the best for the datasets and models we evaluated. With FreeLB, we achieve state-of-the-art results on several important language understanding benchmarks. On the GLUE benchmark, FreeLB pushes the performance of the BERT-base model from 78.3 to 79.4. The overall score of the RoBERTa-large models on the GLUE benchmark is also lifted from 88.5 to 88.8, achieving best results on most of its sub-tasks. Experiments also show that FreeLB can boost the performance of RoBERTa-large on question answering tasks, such as the ARC and CommonsenseQA benchmarks. We also provide a comprehensive ablation study and analysis to demonstrate the effectiveness of our training process.

## 2 Related Work

### 2.1 Adversarial Training

To improve the robustness of neural networks against adversarial examples, many defense strategies and models have been proposed, in which PGD-based adversarial training (Madry et al., 2018) is widely considered to be the most effective, since it largely avoids the the obfuscated gradient problem (Athalye et al., 2018). It formulates a class of adversarial training algorithms (Kurakin et al., 2017) into solving a minimax problem on the cross-entropy loss, which can be achieved reliably through multiple projected gradient ascent steps followed by a SGD (Stochastic Gradient Descent) step.

Despite being verified by Athalye et al. (2018) to avoid obfuscated gradients, Qin et al. (2019) shows that PGD-based adversarial training still leads to highly convoluted and non-linear loss surfaces when $K$ is small, which could be readily broken under stronger adversaries. Thus, to be effective, the cost of PGD-based adversarial training is much higher than conventional training. To mitigate this cost, Shafahi et al. (2019) proposed a "free" adversarial training algorithm that simultaneously updates both model parameters and adversarial perturbations on a single backward pass. Using a similar formulation, Zhang et al. (2019) effectively reduce the total number of full forward and backward propagations for obtaining adversarial examples by restricting most of its adversarial updates in the first layer.

### 2.2 Adversarial Examples in Natural Languages

Adversarial examples have been explored primarily in the image domain, and received many attention in text domain recently. Previous works on text adversaries have focused on heuristics for creating adversarial examples in the black-box setting, or on specific tasks. Jia & Liang (2017) propose to add distracting sentences to the input document in order to induce mis-classification. Zhao et al. (2018) generate text adversaries by projecting the input data to a latent space using GANs, and searching for adversaries close to the original instance. Belinkov & Bisk (2018) manipulate every word in a sentence with synthetic or natural noise in machine translation systems. Iyyer et al. (2018) propose a neural paraphrase model based on back-translated data to produce paraphrases that have different sentence structures. Different from previous work, ours is not to produce actual adversarial examples, but only take the benefit of adversarial training for natural language understanding.

We are not the first to observe that robust language models may perform better on clean test data. Miyato et al. (2017) extend adversarial and virtual adversarial training (Miyato et al., 2019) to the text domain to improve the performance on semi-supervised classification tasks. Ebrahimi et al. (2018) propose a character/word replacement for crafting attacks, and show employing adversarial examples in training renders the models more robust. Ribeiro et al. (2018) show that adversarial attacks can be used as a valuable tool for debugging NLP models. Cheng et al. (2019) also find that crafting adversarial examples can help neural machine translation significantly. Notably, these studies have focused on simple models or text generation tasks. Our work explores how to efficiently use the gradients obtained in adversarial training to boost the performance of state-of-the-art transformer-based models.

# 3 ADVERSARIAL TRAINING FOR LANGUAGE UNDERSTANDING

Pre-trained large-scale language models, such as BERT (Devlin et al., 2019), RoBERTa (Liu et al., 2019b), ALBERT (Lan et al., 2020) and T5 (Raffel et al., 2019), have proven to be highly effective for downstream tasks. We aim to further improve the generalization of these pre-trained language models on the downstream language understanding tasks by enhancing their robustness in the embedding space during finetuning on these tasks. We achieve this goal by creating "virtual" adversarial examples in the embedding space, and then perform parameter updates on these adversarial embeddings. Creating actual adversarial examples for language is difficult; even with state-of-the-art language models as guidance (e.g., (Cheng et al., 2019)), it remains unclear how to construct label-preserving adversarial examples via word/character replacement without human evaluations, because the meaning of each word/character depends on the context (Ribeiro et al., 2018). Since we are only interested in the *effects* of adversarial training, rather than producing actual adversarial examples, we add norm-bounded adversarial perturbations to the embeddings of the input sentences using a gradient-based method. Note that our embedding-based adversary is strictly stronger than a more conventional text-based adversary, as our adversary can make manipulations on word embeddings that are not possible in the text domain.

For models that incorporate various input representations, including word or subword embeddings, segment embeddings and position embeddings, our adversaries only modify the concatenated word or sub-word embeddings, leaving other components of the sentence representation unchanged. [2] Denote the sequence of one-hot representations of the input subwords as $\boldsymbol{Z} = [\boldsymbol{z}_1, \boldsymbol{z}_2, ..., \boldsymbol{z}_n]$, the embedding matrix as $\boldsymbol{V}$, and the language model (encoder) as a function $\boldsymbol{y} = f_{\boldsymbol{\theta}}(\boldsymbol{X})$, where $\boldsymbol{X} = \boldsymbol{V}\boldsymbol{Z}$ is the subword embeddings, $\boldsymbol{y}$ is the output of the model (e.g., class probabilities for classification models), and $\boldsymbol{\theta}$ denotes all the learnable parameters including the embedding matrix $\boldsymbol{V}$. We add adversarial perturbations $\boldsymbol{\delta}$ to the embeddings such that the prediction becomes $\boldsymbol{y}' = f_{\boldsymbol{\theta}}(\boldsymbol{X} + \boldsymbol{\delta})$. To preserve the semantics, we constrain the norm of $\boldsymbol{\delta}$ to be small, and assume the model's prediction should not change after the perturbation. This formulation is analogous to Miyato et al. (2017), with the difference that we do not require $\boldsymbol{X}$ to be normalized.

## 3.1 PGD FOR ADVERSARIAL TRAINING

Standard adversarial training seeks to find optimal parameters $\boldsymbol{\theta}^*$ to minimize the maximum risk for any $\boldsymbol{\delta}$ within a norm ball as:

$$\min_{\boldsymbol{\theta}} \mathbb{E}_{(\boldsymbol{Z},y)\sim\mathcal{D}} \left[ \max_{\|\boldsymbol{\delta}\|\leq\epsilon} L(f_{\boldsymbol{\theta}}(\boldsymbol{X} + \boldsymbol{\delta}), y) \right], \tag{1}$$

where $\mathcal{D}$ is the data distribution, $y$ is the label, and $L$ is some loss function. We use the Frobenius norm to constrain $\boldsymbol{\delta}$. For neural networks, the outer "min" is non-convex, and the inner "max" is non-concave. Nonetheless, Madry et al. (2018) demonstrated that this saddle-point problem can be solved reliably with SGD for the outer minimization and PGD (a standard method for large-scale constrained optimization, see (Combettes & Pesquet, 2011) and (Goldstein et al., 2014)), for the inner maximization. In particular, for the constraint $\|\boldsymbol{\delta}\|_F \leq \epsilon$, with an additional assumption that the loss function is locally linear, PGD takes the following step (with step size $\alpha$) in each iteration:

$$\boldsymbol{\delta}_{t+1} = \Pi_{\|\boldsymbol{\delta}\|_F \leq \epsilon} \left( \boldsymbol{\delta}_t + \alpha g(\boldsymbol{\delta}_t)/\|g(\boldsymbol{\delta}_t)\|_F \right), \tag{2}$$

where $g(\boldsymbol{\delta}_t) = \nabla_{\boldsymbol{\delta}} L(f_{\boldsymbol{\theta}}(\boldsymbol{X} + \boldsymbol{\delta}_t), y)$ is the gradient of the loss with respect to $\boldsymbol{\delta}$, and $\Pi_{\|\boldsymbol{\delta}\|_F \leq \epsilon}$ performs a projection onto the $\epsilon$-ball. To achieve high-level robustness, multi-step adversarial examples are needed during training, which is computationally expensive. The $K$-step PGD ($K$-PGD) requires $K$ forward-backward passes through the network, while the standard SGD update requires only one. As a result, the adversary generation step in adversarial training increases run-time by an order of magnitude—a catastrophic amount when training large state-of-the-art language models.

## 3.2 LARGE-BATCH ADVERSARIAL TRAINING FOR FREE

In the inner ascent steps of PGD, the gradients of the parameters can be obtained with almost no overhead when computing the gradients of the inputs. From this observation, FreeAT (Shafahi et al.,

---

[2]"Subword embeddings" refers to the embeddings of sub-word encodings such as the popular Byte Pair Encoding (BPE) (Sennrich et al., 2016).

---

**Algorithm 1** "Free" Large-Batch Adversarial Training (FreeLB-$K$)

---

**Require:** Training samples $X = \{(\boldsymbol{Z}, y)\}$, perturbation bound $\epsilon$, learning rate $\tau$, ascent steps $K$, ascent step size $\alpha$
1: Initialize $\boldsymbol{\theta}$
2: **for** epoch $= 1 \ldots N_{ep}$ **do**
3:     **for** minibatch $B \subset X$ **do**
4:         $\boldsymbol{\delta}_0 \leftarrow \frac{1}{\sqrt{N_\delta}} U(-\epsilon, \epsilon)$
5:         $\boldsymbol{g}_0 \leftarrow 0$
6:         **for** $t = 1 \ldots K$ **do**
7:             Accumulate gradient of parameters $\theta$
8:                 $\boldsymbol{g}_t \leftarrow \boldsymbol{g}_{t-1} + \frac{1}{K} \mathbb{E}_{(\boldsymbol{Z},y) \in B}[\nabla_{\boldsymbol{\theta}} L(f_{\boldsymbol{\theta}}(\boldsymbol{X} + \boldsymbol{\delta}_{t-1}), y)]$
9:             Update the perturbation $\delta$ via gradient ascend
10:                 $\boldsymbol{g}_{adv} \leftarrow \nabla_{\boldsymbol{\delta}} L(f_{\boldsymbol{\theta}}(\boldsymbol{X} + \boldsymbol{\delta}_{t-1}), y)$
11:                 $\boldsymbol{\delta}_t \leftarrow \Pi_{\|\boldsymbol{\delta}\|_F \le \epsilon}(\boldsymbol{\delta}_{t-1} + \alpha \cdot \boldsymbol{g}_{adv} / \|\boldsymbol{g}_{adv}\|_F)$
12:         **end for**
13:         $\boldsymbol{\theta} \leftarrow \boldsymbol{\theta} - \tau \boldsymbol{g}_K$
14:     **end for**
15: **end for**

---

2019) and YOPO (Zhang et al., 2019) have been proposed to accelerate adversarial training. They achieve comparable robustness and generalization as standard PGD-trained models using only the same or a slightly larger number of forward-backward passes as natural training (i.e., SGD on clean samples). FreeAT takes one descent step on the parameters together with *each* of the $K$ ascent steps on the perturbation. As a result, FreeAT may suffer from the "stale gradient" problem (Dutta et al., 2018), where in every step $t$, $\boldsymbol{\delta}_t$ does not necessarily maximize the model with parameter $\boldsymbol{\theta}_t$ since its update is based on $\nabla_{\boldsymbol{\delta}} L(f_{\boldsymbol{\theta}_{t-1}}(\boldsymbol{X} + \boldsymbol{\delta}_{t-1}), y)$, and vice versa, $\boldsymbol{\theta}_t$ does not necessarily minimize the adversarial risk with adversary $\boldsymbol{\delta}_t$ since its update is based on $\nabla_{\boldsymbol{\theta}} L(f_{\boldsymbol{\theta}_{t-1}}(\boldsymbol{X} + \boldsymbol{\delta}_{t-1}), y)$. Such a problem may be more significant when the step size is large.

Different from FreeAT, YOPO accumulates the gradient of the parameters from each of the ascent steps, and updates the parameters only once after the $K$ inner ascent steps. YOPO also advocates that after each back-propagation, one should take the gradient of the first hidden layer as a constant and perform several additional updates on the adversary using the product of this constant and the Jacobian of the first layer of the network to obtain strong adversaries. However, when the first hidden layer is a linear layer as in their implementation, such an operation is equivalent to taking a larger step size on the adversary. The analysis backing the extra update steps also assumes a twice continuously differentiable loss, which does not hold for ReLU-based neural networks they experimented with, and thus the reasons for the success of such an algorithm remains obscure. We give empirical comparisons between YOPO and our approach in Sec. 4.3.

To obtain better solutions for the inner max and avoid fundamental limitations on the function class, we propose FreeLB, which performs multiple PGD iterations to craft adversarial examples, and simultaneously accumulates the "free" parameter gradients $\nabla_{\boldsymbol{\theta}} L$ in each iteration. After that, it updates the model parameter $\boldsymbol{\theta}$ all at once with the accumulated gradients. The overall procedure is shown in Algorithm 1, in which $\boldsymbol{X} + \boldsymbol{\delta}_t$ is an approximation to the local maximum within the intersection of two balls $\mathcal{I}_t = \mathcal{B}_{\boldsymbol{X}+\boldsymbol{\delta}_0}(\alpha t) \cap \mathcal{B}_{\boldsymbol{X}}(\epsilon)$. By taking a descent step along the averaged gradients at $\boldsymbol{X} + \boldsymbol{\delta}_0, ..., \boldsymbol{X} + \boldsymbol{\delta}_{K-1}$, we approximately optimize the following objective:

$$\min_{\boldsymbol{\theta}} \mathbb{E}_{(\boldsymbol{Z},y) \sim \mathcal{D}} \left[ \frac{1}{K} \sum_{t=0}^{K-1} \max_{\boldsymbol{\delta}_t \in \mathcal{I}_t} L(f_{\boldsymbol{\theta}}(\boldsymbol{X} + \boldsymbol{\delta}_t), y) \right], \tag{3}$$

which is equivalent to replacing the original batch $\boldsymbol{X}$ with a $K$-times larger virtual batch, consisting of samples whose embeddings are $\boldsymbol{X} + \boldsymbol{\delta}_0, ..., \boldsymbol{X} + \boldsymbol{\delta}_{K-1}$. Compared with PGD-based adversarial training (Eq. 1), which minimizes the maximum risk at a single estimated point in the vicinity of each training sample, FreeLB minimizes the maximum risk at each ascent step at almost no overhead.

Intuitively, FreeLB could be a learning method with lower generalization error than PGD. Sokolic et al. (2017) have proved that the generalization error of a learning method invariant to a set of $T$ transformations may be up to $\sqrt{T}$ smaller than a non-invariant learning method. According to

their theory, FreeLB could have a more significant improvement over natural training, since FreeLB enforces the invariance to $K$ adversaries from a set of up to $K$ different norm constraints,[3] while PGD only enforces invariance to a single norm constraint $\epsilon$.

Empirically, FreeLB does lead to higher robustness and invariance than PGD in the embedding space, in the sense that the maximum increase of loss in the vicinity of $\boldsymbol{X}$ for models trained with FreeLB is smaller than that with PGD. See Sec. 4.3 for details. In theory, such improved robustness can lead to better generalization (Xu & Mannor, 2012), which is consistent with our experiments. Qin et al. (2019) also demonstrated that PGD-based method leads to highly convoluted and non-linear loss surfaces in the vicinity of input samples when $K$ is small, indicating a lack of robustness.

### 3.3 WHEN ADVERSARIAL TRAINING MEETS DROPOUT

Usually, adversarial training is not used together with dropout (Srivastava et al., 2014). However, for some language models like RoBERTa (Liu et al., 2019b), dropout is used during the finetuning stage. In practice, when dropout is turned on, each ascent step of Algorithm 1 is optimizing $\boldsymbol{\delta}$ for a different network. Specifically, denote the dropout mask as $\boldsymbol{m}$ with each entry $m_i \sim \text{Bernoulli}(p)$. Similar to our analysis for FreeAT, the ascent step from $\boldsymbol{\delta}_{t-1}$ to $\boldsymbol{\delta}_t$ is based on $\nabla_{\boldsymbol{\delta}} L(f_{\boldsymbol{\theta}(\boldsymbol{m}_{t-1})}(\boldsymbol{X} + \boldsymbol{\delta}_{t-1}), y)$, so $\boldsymbol{\delta}_t$ is sub-optimal for $L(f_{\boldsymbol{\theta}(\boldsymbol{m}_t)}(\boldsymbol{X} + \boldsymbol{\delta}), y)$. Here $\boldsymbol{\theta}(\boldsymbol{m})$ is the effective parameters under dropout mask $\boldsymbol{m}$.

The more plausible solution is to use the same $\boldsymbol{m}$ in each step. When applying dropout to any network, the objective for $\boldsymbol{\theta}$ is to minimize the expectation of loss under different networks determined by the dropout masks, which is achieved by minimizing the Monte Carlo estimation of the expected loss. In our case, the objective becomes:

$$\min_{\boldsymbol{\theta}} \mathbb{E}_{(\boldsymbol{Z},\boldsymbol{y})\sim\mathcal{D},\boldsymbol{m}\sim\mathcal{M}} \left[ \frac{1}{K} \sum_{t=0}^{K-1} \max_{\boldsymbol{\delta}_t\in\mathcal{I}_t} L(f_{\boldsymbol{\theta}(\boldsymbol{m})}(\boldsymbol{X} + \boldsymbol{\delta}_t), y) \right],\tag{4}$$

where the 1-sample Monte Carlo estimation should be $\frac{1}{K}\sum_{t=0}^{K-1}\max_{\boldsymbol{\delta}_t\in\mathcal{I}_t} L(f_{\boldsymbol{\theta}(\boldsymbol{m}_0)}(\boldsymbol{X} + \boldsymbol{\delta}_t), y)$ and can be minimized by using FreeLB with dropout mask $\boldsymbol{m}_0$ in each ascent step. This is similar to applying Variational Dropout to RNNs as used in Gal & Ghahramani (2016).

## 4 EXPERIMENTS

In this section, we provide comprehensive analysis on FreeLB through extensive experiments on three Natural Language Understanding benchmarks: GLUE (Wang et al., 2019), ARC (Clark et al., 2018) and CommonsenseQA (Talmor et al., 2019). We also compare the robustness and generalization of FreeLB with other adversarial training algorithms to demonstrate its strength. Additional experimental details are provided in the Appendix.

### 4.1 DATASETS

**GLUE Benchmark.** The GLUE benchmark is a collection of 9 natural language understanding tasks, namely Corpus of Linguistic Acceptability (CoLA; Warstadt et al. (2018)), Stanford Sentiment Treebank (SST; Socher et al. (2013)), Microsoft Research Paraphrase Corpus (MRPC; Dolan & Brockett (2005)), Semantic Textual Similarity Benchmark (STS; Agirre et al. (2007)), Quora Question Pairs (QQP; Iyer et al. (2017)), Multi-Genre NLI (MNLI; Williams et al. (2018)), Question NLI (QNLI; Rajpurkar et al. (2016)), Recognizing Textual Entailment (RTE; Dagan et al. (2006); Bar Haim et al. (2006); Giampiccolo et al. (2007); Bentivogli et al. (2009)) and Winograd NLI (WNLI; Levesque et al. (2011)). 8 of the tasks are formulated as classification problems and only STS-B is formulated as regression, but FreeLB applies to all of them. For BERT-base, we use the HuggingFace implementation[4], and follow the single-task finetuning procedure as in Devlin et al. (2019). For RoBERTa, we use the fairseq implementation[5]. Same as Liu et al. (2019b), we also use

---

[3]The cardinality of the set is approximately $\min\{K, \lceil \frac{\epsilon - \mathbb{E}[\|\delta_0\|]}{\alpha} \rceil + 1\}$.

[4]https://github.com/huggingface/pytorch-transformers

[5]https://github.com/pytorch/fairseq

| Method | MNLI (Acc) | QNLI (Acc) | QQP (Acc) | RTE (Acc) | SST-2 (Acc) | MRPC (Acc) | CoLA (Mcc) | STS-B (Pearson) |
|---|---|---|---|---|---|---|---|---|
| Reported | 90.2 | 94.7 | 92.2 | 86.6 | 96.4 | 90.9 | 68.0 | 92.4 |
| ReImp | - | - | - | 85.61 (1.7) | 96.56 (.3) | 90.69 (.5) | 67.57 (1.3) | 92.20 (.2) |
| PGD | 90.53 (.2) | 94.87 (.2) | 92.49 (.07) | 87.41 (.9) | 96.44 (.1) | 90.93 (.2) | 69.67 (1.2) | 92.43 (7.) |
| FreeAT | 90.02 (.2) | 94.66 (.2) | 92.48 (.08) | 86.69 (15.) | 96.10 (.2) | 90.69 (.4) | 68.80 (1.3) | 92.40 (.3) |
| FreeLB | **90.61** (.1) | **94.98** (.2) | **92.60** (.03) | **88.13** (1.2) | **96.79** (.2) | **91.42** (.7) | **71.12** (.9) | **92.67** (.08) |

Table 1: Results (median and variance) on the dev sets of GLUE based on the RoBERTa-large model, from 5 runs with the same hyperparameter but different random seeds. ReImp is our reimplementation of RoBERTa-large. The training process can be very unstable even with the vanilla version. Here, both PGD on STS-B and FreeAT on RTE demonstrates such instability, with one unconverged instance out of five.

| Model | Score | CoLA 8.5k | SST-2 67k | MRPC 3.7k | STS-B 7k | QQP 364k | MNLI-m/mm 393k | QNLI 108k | RTE 2.5k | WNLI 634 | AX |
|---|---|---|---|---|---|---|---|---|---|---|---|
| BERT-base[1] | 78.3 | 52.1 | 93.5 | 88.9/84.8 | 87.1/85.8 | 71.2/89.2 | 84.6/83.4 | 90.5 | 66.4 | 65.1 | 34.2 |
| FreeLB-BERT | 79.4 | 54.5 | 93.6 | 88.1/83.5 | 87.7/86.7 | 72.7/89.6 | 85.7/84.6 | 91.8 | 70.1 | 65.1 | 36.9 |
| MT-DNN[2] | 87.6 | **68.4** | 96.5 | 92.7/90.3 | 91.1/90.7 | 73.7/89.9 | 87.9/87.4 | 96.0 | 86.3 | 89.0 | 42.8 |
| XLNet-Large[3] | 88.4 | 67.8 | **96.8** | 93.0/90.7 | 91.6/91.1 | 74.2/90.3 | 90.2/89.8 | 98.6 | 86.3 | **90.4** | 47.5 |
| RoBERTa[4] | 88.5 | 67.8 | 96.7 | 92.3/89.8 | 92.2/91.9 | 74.3/90.2 | 90.8/90.2 | **98.9** | 88.2 | 89.0 | 48.7 |
| FreeLB-RoB | **88.8** | 68.0 | **96.8** | **93.1/90.8** | **92.4/92.2** | **74.8/90.3** | **91.1/90.7** | 98.8 | **88.7** | 89.0 | **50.1** |
| Human | 87.1 | 66.4 | 97.8 | 86.3/80.8 | 92.7/92.6 | 59.5/80.4 | 92.0/92.8 | 91.2 | 93.6 | 95.9 | - |

Table 2: Results on GLUE from the evaluation server, as of Sep 25, 2019. Metrics are the same as the leaderboard. Number under each task's name is the size of the training set. FreeLB-BERT is the single-model results of BERT-base finetuned with FreeLB, and FreeLB-RoB is the ensemble of 7 RoBERTa-Large models for each task. References: [1]: (Devlin et al., 2019); [2]: (Liu et al., 2019a); [3]: (Yang et al., 2019); [4]: (Liu et al., 2019b).

single-task finetuning for all dev set results, and start with MNLI-finetuned models on RTE, MRPC and STS-B for the test submissions.

**ARC Benchmark.** The ARC dataset (Clark et al., 2018) is a collection of multi-choice science questions from grade-school level exams. It is further divided into ARC-Challenge set with 2,590 question answer (QA) pairs and ARC-Easy set with 5,197 QA pairs. Questions in ARC-Challenge are more difficult and cannot be handled by simply using a retrieval and co-occurence based algorithm (Clark et al., 2018). A typical question is:

*Which property of a mineral can be determined just by looking at it?*

*(A) luster [correct] (B) mass (C) weight (D) hardness.*

**CommonsenseQA Benchmark.** The CommonsenseQA dataset (Talmor et al., 2019) consists of 12,102 natural language questions that require human commonsense reasoning ability to answer. A typical question is :

*Where can I stand on a river to see water falling without getting wet?*

*(A) waterfall, (B) bridge [correct], (C) valley, (D) stream, (E) bottom.*

Each question has five candidate answers from ConceptNet (Speer et al., 2017). To make the question more difficult to solve, most answers have the same relation in ConceptNet to the key concept in the question. As shown in the above example, most answers can be connected to "river" by "At-Location" relation in ConceptNet. For a fair comparison with the reported results in papers and leaderboard[6], we use the official random split 1.11.

### 4.2 EXPERIMENTAL RESULTS

**GLUE** We summarize results on the dev sets of GLUE in Table 1, comparing the proposed FreeLB against other adversatial training algorithms (PGD (Madry et al., 2018) and FreeAT (Shafahi et al., 2019)). We use the same step size $\alpha$ and number of steps $m$ for PGD, FreeAT and FreeLB. FreeLB is consistently better than the two baselines. Comparisons and detailed discussions about

---

[6]https://www.tau-nlp.org/csqa-leaderboard

| | ARC-Easy | | ARC-Challenge | | ARC-Merge | | CQA | | |
| --- | --- | --- | --- | --- | --- | --- | --- | --- | --- |
| | Dev | Test | Dev | Test | Dev | Test | Dev | Test | Test (E) |
| RoBERTa (Reported) | - | - | - | - | - | - | 78.43 | 72.1 | 72.5 |
| RoBERTa (ReImp) | 84.39 | 84.13 | 64.54 | 64.44 | 77.83 | 77.62 | 77.56 | - | - |
| FreeLB-RoBERTa | **84.91** | 84.81 | 65.89 | 65.36 | 78.37 | 78.39 | **78.81** | **72.2** | **73.1** |
| AristoRoBERTaV7 (MTL) | - | 85.02 | - | 66.47 | - | 78.89 | - | - | - |
| XLNet + RoBERTa (MTL+Ens) | - | - | - | 67.06 | - | - | - | - | - |
| FreeLB-RoBERTa (MTL) | **84.91** | **85.44** | **70.23** | **67.75** | **79.86** | **79.60** | - | - | - |

Table 3: Results on ARC and CommonsenseQA (CQA). ARC-Merge is the combination of ARC-Easy and ARC-Challenge, "MTL" stands for multi-task learning and "Ens" stands for ensemble. Results of XLNet + RoBERTa (MTL+Ens) and AristoRoBERTaV7 (MTL) are from the ARC leaderboards. Test (E) denotes the test set results with ensembles. For CQA, we report the highest dev and test accuracies among *all* models. The models with 78.81/72.19 dev/test accuracy (as in the table) have 71.84/78.64 test/dev accuracies respectively.

YOPO (Zhang et al., 2019) are provided in Sec. 4.3. We have also submitted our results to the evaluation server, results provided in Table 2. FreeLB lifts the performance of the BERT-base model from 78.3 to 79.4, and RoBERTa-large model from 88.5 to 88.8 on overall scores.

**ARC** For ARC, a corpus of 14 million related science documents (from ARC Corpus, Wikipedia and other sources) is provided. For each QA pair, we first use a retrieval model to select top 10 related documents. Then, given these retrieved documents[7], we use RoBERTa-large model to encode $\langle s \rangle$ *Retrieved Documents* $\langle /s \rangle$ *Question + Answer* $\langle /s \rangle$, where $\langle s \rangle$ and $\langle /s \rangle$ are special tokens for RoBERTa model[8]. We then apply a fully-connected layer to the representation of the [CLS] token to compute the final logit, and use standard cross-entropy loss for model training.

Results are summarized in Table 3. Following Sun et al. (2018), we first finetune the RoBERTa model on the RACE dataset (Lai et al., 2017). The finetuned RoBERTa model achieves 85.70% and 85.24% accuracy on the development and test set of RACE, respectively. Based on this, we further finetune the model on both ARC-Easy and ARC-Challenge datasets with the same hyper-parameter searching strategy (for 5 epochs), which achieves 84.13%/64.44% test accuracy on ARC-Easy/ARC-Challenge. And by adding FreeLB finetuning, we can reach 84.81%/65.36%, a significant boost on ARC benchmark, demonstrating the effectiveness of FreeLB.

To further improve the results, we apply a multi-task learning (MTL) strategy using additional datasets. We first finetune the model on RACE (Lai et al., 2017), and then finetune on a joint dataset of ARC-Easy, ARC-Challenge, OpenbookQA (Mihaylov et al., 2018) and Regents Living Environment[9]. Based on this, we further finetune our model on ARC-Easy and ARC-Challenge with FreeLB. After finetuning, our single model achieves 67.75% test accuracy on ARC-Challenge and 85.44% on ARC-Easy, both outperforming the best submission on the official leaderboard[10].

**CommonsenseQA** Similar to the training strategy in Liu et al. (2019b), we construct five inputs for each question by concatenating the question and each answer separately, then encode each input with the representation of the [CLS] token. A final score is calculated by applying the representation of [CLS] to a fully-connected layer. Following the fairseq repository[11], the input is formatted as: "$\langle s \rangle$ *Q: Where can I stand on a river to see water falling without getting wet?* $\langle /s \rangle$ *A: waterfall* $\langle /s \rangle$", where '*Q:*' and '*A:*' are the prefix for question and answer, respectively.

Results are summarized in Table 3. We obtained a dev-set accuracy of 77.56% with the RoBERTa-large model. When using FreeLB finetuning, we achieved 78.81%, a 1.25% absolute gain. Compared with the results reported from fairseq repository, which obtains 78.43% accuracy on the dev-set, FreeLB still achieves better performance. Our submission to the CommonsenseQA leaderboard achieves 72.2% single-model test set accuracy, and the result of a 20-model ensemble is 73.1%, which achieves No.1 among all the submissions without making use of ConceptNet.

---

[7]We thank AristoRoBERTa team for providing retrieved documents and additional Regents Living Environments dataset.

[8]Equivalent to [CLS] and [SEP] token in BERT.

[9]https://www.nysedregents.org/livingenvironment

[10]https://leaderboard.allenai.org/arc/submissions/public and https://leaderboard.allenai.org/arc_easy/submissions/public

[11]https://github.com/pytorch/fairseq/tree/master/examples/roberta/commonsense_qa

| Methods | Vanilla | FreeLB-3* | FreeLB-3 | YOPO-3-2 | YOPO-3-3 |
|---------|---------|-----------|----------|----------|----------|
| RTE | 85.61 (1.67) | 87.14 (1.29) | **88.13 (1.21)** | 87.05 (1.36) | 87.05 (0.20) |
| CoLA | 67.57 (1.30) | 69.31 (1.16) | **71.12 (0.90)** | 70.40 (0.91) | 69.91 (1.16) |
| MRPC | 90.69 (0.54) | 90.93 (0.66) | **91.42 (0.72)** | 90.44 (0.62) | 90.69 (0.37) |

Table 4: The median and standard deviation of the scores on the dev sets of RTE, CoLA and MRPC from the GLUE benchmark, computed from 5 runs with the same hyper-parameters except for the random seeds. We use FreeLB-$m$ to denote FreeLB with $m$ ascent steps, and FreeLB-3* to denote the version without reusing the dropout mask.

| Methods | RTE | | | CoLA | | | MRPC | | |
|---------|-----|-----|-----|------|------|------|------|------|------|
| | M-Inc $(10^{-4})$ | M-Inc (R) $(10^{-4})$ | N-Loss $(10^{-4})$ | M-Inc $(10^{-4})$ | M-Inc (R) $(10^{-4})$ | N-Loss $(10^{-4})$ | M-Inc $(10^{-3})$ | M-Inc (R) $(10^{-3})$ | N-Loss $(10^{-3})$ |
| Vanilla | 5.1 | 5.3 | 4.5 | 6.1 | 5.7 | 5.2 | 10.2 | 10.2 | 1.9 |
| PGD | 4.7 | 4.9 | 6.2 | 128.2 | 130.1 | 436.1 | 5.7 | 5.7 | 5.4 |
| FreeLB | 3.0 | 2.6 | 4.1 | 1.4 | 1.3 | 7.2 | 3.6 | 3.6 | 2.7 |

Table 5: Median of the maximum increase in loss in the vicinity of the dev set samples for RoBERTa-Large model finetuned with different methods. Vanilla models are naturally trained RoBERTa's. M-Inc: Max Inc, M-Inc (R): Max Inc (R). Nat Loss (N-Loss) is the loss value on clean samples. Notice we require *all* clean samples here to be correctly classified by all models, which results in 227, 850 and 355 samples for RTE, CoLA and MRPC, respectively. We also give the variance in the Appendix.

## 4.3 ABLATION STUDY AND ANALYSIS

In this sub-section, we first show the importance of reusing dropout mask, then conduct a thorough ablation study on FreeLB over the GLUE benchmark to analyze the robustness and generalization strength of different approaches. We observe that it is unnecessary to perform shallow-layer updates on the adversary as YOPO for our case, and FreeLB results in improved robustness and generalization compared with PGD.

**Importance of Reusing Mask** Table 4 (columns 2 to 4) compares the results of FreeLB with and without reusing the same dropout mask in each ascent step, as proposed in Sec. 3.3. With reusing, FreeLB can achieve a larger improvement over the naturally trained models. Thus, we enable mask reusing for all experiments involving RoBERTa.

**Comparing the Robustness** Table 5 provides the comparisons of the maximum increment of loss in the vicinity of each sample, defined as:

$$\Delta L_{\max}(\boldsymbol{X}, \epsilon) = \max_{\|\boldsymbol{\delta}\| \leq \epsilon} L(f_{\boldsymbol{\theta}}(\boldsymbol{X} + \boldsymbol{\delta}), y) - L(f_{\boldsymbol{\theta}}(\boldsymbol{X}), y), \tag{5}$$

which reflects the robustness and invariance of the model in the embedding space. In practice, we use PGD steps as in Eq. 2 to find the value of $\Delta L_{\max}(\boldsymbol{X}, \epsilon)$. We found that when using a step size of $5 \cdot 10^{-3}$ and $\epsilon = 0.01\|X\|_F$, the PGD iterations converge to almost the same value, starting from 100 different random initializations of $\boldsymbol{\delta}$ for the RoBERTa models, trained with or without FreeLB. This indicates that PGD reliably finds $\Delta L_{\max}$ for these models. Therefore, we compute $\Delta L_{\max}(\boldsymbol{X}, \epsilon)$ for each $\boldsymbol{X}$ via a 2000-step PGD.

Samples with small margins exist even for models with perfect accuracy, which could give a false sense of vulnerability of the model. To rule out the outlier effect and make $\Delta L_{\max}(\boldsymbol{X}, \epsilon)$ comparable across different samples, we only consider samples that all the evaluated models can correctly classify, and search for an $\epsilon$ for each sample such that the reference model can correctly classify all samples within the $\epsilon$ ball.[12] However, such choice of per-sample $\epsilon$ favors the reference model by design. To make fair comparisons, Table 5 provides the median of $\Delta L_{\max}(\boldsymbol{X}, \epsilon)$ with per-sample $\epsilon$ from models trained by FreeLB (Max Inc) and PGD (Mac Inc (R)), respectively.

Across all three datasets and different reference models, FreeLB has the smallest median increment even when starting from a larger natural loss than vanilla models. This demonstrates that FreeLB is more robust and invariant in most cases. Such results are also consistent with the models' dev set performance (the performances for Vanilla/PGD/FreeLB models on RTE, CoLA and MRPC are 86.69/87.41/89.21, 69.91/70.84/71.40, 91.67/91.17/91.17, respectively).

---

[12]For each sample, we start from a value slightly larger than the norm constraint during training for $\epsilon$, and then decrease $\epsilon$ linearly until the model trained with the reference model can correctly classify after a 2000-step PGD attack. The reference model is either trained with FreeLB or PGD.

**Comparing with YOPO** The original implementation of YOPO (Zhang et al., 2019) chooses the first convolutional layer of the ResNets as $f_0$ for updating the adversary in the "s-loop". As a result, each step of the "s-loop" should be using exactly the same value to update the adversary, and YOPO-$m$-$n$ degenerates into FreeLB with a $n$-times large step size. To avoid that, we choose the layers up to the output of the first Transformer block as $f_0$ when implementing YOPO. To make the total amount of update on the adversary equal, we take the hyper-parameters for FreeLB-$m$ and only change the step size $\alpha$ into $\alpha/n$ for YOPO-$m$-$n$. Table 4 shows that FreeLB performs consistently better than YOPO on all three datasets. Accidentally, we also give the results comparing with YOPO-$m$-$n$ *without* changing the step size $\alpha$ for YOPO in Table 8. The gap between two approaches seem to shrink, which may be caused by using a larger total step size for the YOPO adversaries. We leave exhaustive hyperparameter search for both models as our future work.

## 5 CONCLUSION

In this work, we have developed an adversarial training approach, FreeLB, to improve natural language understanding. The proposed approach adds perturbations to continuous word embeddings using a gradient method, and minimizes the resultant adversarial risk in an efficient way. FreeLB is able to boost Transformer-based model (BERT and RoBERTa) on several datasets and achieve new state of the art on GLUE and ARC benchmarks. Empirical study demonstrates that our method results in both higher robustness in the embedding space than natural training and better generalization ability. Such observation is also consistent with recent findings in Computer Vision. However, adversarial training still takes significant overhead compared with vanilla SGD. How to accelerate this process while improving generalization is an interesting future direction.

**Acknowledgements:** Goldstein and Zhu were supported in part by the DARPA GARD, DARPA QED for RML, and AFOSR MURI programs.

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

# A    ADDITIONAL EXPERIMENTAL DETAILS

## A.1    PROBLEM FORMULATIONS

For tasks with ranking loss like ARC, CommonsenseQA, WNLI and QNLI, add the perturbation to the concatenation of the embeddings of all question/answer pairs.

Additional tricks are required to achieve high performance on WNLI and QNLI for the GLUE benchmark. We use the same tricks as Liu et al. (2019b). For WNLI, we use the same WSC data provided by Liu et al. (2019b) for training. For testing, Liu et al. (2019b) also provided the test set with span annotations, but the order is different form the GLUE dataset. We re-order their test set by matching. For the QNLI, we follow Liu et al. (2019b) and formulate the problem as pairwise ranking problem, which is the same for CommonsenseQA. We find the matching pairs for both training set and testing set by matching the queries in the dev set. We predict "entailment" if the candidate has the higher score, and "not_entailment" otherwise.

## A.2    HYPER-PARAMETERS

|   | MNLI | QNLI | QQP | RTE | SST-2 | MRPC | CoLA | STS-B |
|---|------|------|-----|-----|-------|------|------|-------|
| $\epsilon$ | 2E-1 | 1.5E-1 | 4.5E-1 | 1.5E-1 | 6E-1 | 4E-1 | 2E-1 | 3E-1 |
| $\alpha$ | 1E-1 | 1E-1 | 1.5E-1 | 3E-2 | 1E-1 | 4E-2 | 2.5E-2 | 1E-1 |
| $m$ | 2 | 2 | 2 | 3 | 2 | 3 | 3 | 3 |

Table 6: Additional hyper-parameters on GLUE tasks.

As other adversarial training methods, introduces three additional hyper-parameters: step size $\alpha$, maximum perturbation $\epsilon$, number of steps $m$. For all other hyper-parameters such as learning rate and number of iterations, we either search in the same interval as RoBERTa (on CommonsenseQA, ARC, and WNLI), or use exactly the same setting as RoBERTa (except for MRPC, where we find using a learning rate of $5 \times 10^{-6}$ gives better results).[13] We list the best combinations for $\alpha, \epsilon$ and $m$ for each of the GLUE tasks in Table 6. For WSC/WNLI, the best combination is $\epsilon = 1e - 2, \alpha = 5e - 3, m = 2$. Notice even when $m\alpha < \epsilon$, the maximum perturbation could still reach $\epsilon$ due to the random initialization.

# B    VARIANCE OF MAXIMUM INCREMENT OF LOSS

| Methods | RTE | | | CoLA | | | MRPC | | |
|---------|-----|-----|-----|------|-----|-----|------|-----|-----|
|  | M-Inc $(10^{-4})$ | M-Inc (R) $(10^{-4})$ | N-Loss $(10^{-4})$ | M-Inc $(10^{-4})$ | M-Inc (R) $(10^{-4})$ | N-Loss $(10^{-4})$ | M-Inc $(10^{-3})$ | M-Inc (R) $(10^{-3})$ | N-Loss $(10^{-3})$ |
| Vanilla | 5.1(15087) | 5.3(14346) | 4.5(920) | 6.1(13118) | 5.7(14122) | 5.2(447) | 10.2(929) | 10.2(955) | 1.9(76) |
| PGD | 4.7(10138) | 4.9(1828) | 6.2(752) | 128.2(1300) | 130.1(893) | 436.1(1276) | 5.7(321) | 5.7(155) | 5.4(54) |
| FreeLB | 3.0(1614) | 2.6(11114) | 4.1(595) | 1.4(1392) | 1.3(6231) | 7.2(615) | 3.6(167) | 3.6(601) | 2.7(54) |

Table 7: Median and Standard Deviation of the maximum increase in loss in the vicinity of the dev set samples for RoBERTa-Large model finetuned with different methods. Vanilla models are naturally trained RoBERTa's. Nat Loss is the loss value on clean samples. Notice we require *all* clean samples here to be correctly classified by all models, which results in 227, 850 and 355 samples for RTE, CoLA and MRPC.

Table 7 provides the complete results for the increment of loss in the interval, with median and standard deviation.

# C    ADDITIONAL RESULTS FOR ABLATION STUDIES

Here we provide some additional results for comparison with YOPO as complementary results to Table 4. We will release the complete results for comparing with YOPO and without variational dropout on each of the GLUE tasks in our next revision. From the current results, there is no need

---

[13]https://github.com/pytorch/fairseq/blob/master/examples/roberta/README.glue.md

| Methods | Vanilla | FreeLB-3 | YOPO-3-2 | YOPO-3-3 |
|---------|---------|----------|----------|----------|
| STS-B | 92.20 (.2) | **92.67 (.08)** | 92.60 (.17) | 92.60 (0.20) |
| SST-2 | 96.56 (.3) | **96.79 (.2)** | 96.44 (.2) | 96.33 (.1) |
| QNLI | - | **94.98 (.2)** | 94.96 (.1) | - |
| QQP | - | **92.60 (.03)** | 92.55 (.05)* | 92.50 (.02)* |
| MNLI | - | **90.61 (.1)** | 90.59 (.2) | 90.45 (.2) |

Table 8: The median and standard deviation of the scores on the dev sets of STS-B, SST-2, QNLI, QQP and MNLI from the GLUE benchmark, each computed from 5 runs with the same hyper-parameters except for the random seeds (except for the results with YOPO on QQP, which are from 4 runs). Also note here we use a step size of $\alpha$ for the adversary of YOPO-$m$-$n$, so YOPO effectively uses a step size of $n\alpha$. We use FreeLB-$m$ to denote FreeLB with $m$ ascent steps, and YOPO-3-$n$ to denote YOPO with $n$ shallow-layer ascents.

in using extra shallow-layer updates that YOPO advocates, since this consistently deteriorates the performance while introducing extra computations.

