# OpenReview forum: "FreeLB: Enhanced Adversarial Training for Natural Language Understanding"
_ICLR.cc/2020/Conference — Accept (Spotlight)_

### Official Review · AnonReviewer1 · 2019-10-26
**Official Blind Review #1**

**Rating:** 8

**Review:**


-	This paper modifies and extends the recent “free” training strategies in adversarial training for representation learning for natural language.  The proposed “Free” Large-Batch Adversarial Training is well motived, in comparison with plain PGD-based adversarial training and the existing methods like FreeAT and YOPO, which virtually enlarges the batch size and minimize maximum risk at every ascent step. The contributions are solid.

-	The proposed methods are empirically shown to be effective, in addition to being aligned with some recent theoretic analysis.  The models achieve SOTA on GLUE (by time the paper was submitted; it is not the best model now but that does not affect the contributions), ARC, and the commonsenseQA dataset.

-	The paper conducted good analysis demonstrating the effectiveness of the proposed components, including detailed ablation analysis.

-	The paper is well written. It is well structured and easy to follow.  A minor suggestion (just a personal view) is that the author(s) may consider using “natural  language” instead of just “language” in the title and may consider using more specific words like “representation” instead of “understanding”. But this is minor.

I recommend an accept.


**Experience Assessment:**

I have read many papers in this area.

**Review Assessment: Checking Correctness Of Derivations And Theory:**

I assessed the sensibility of the derivations and theory.

**Review Assessment: Checking Correctness Of Experiments:**

I assessed the sensibility of the experiments.

**Review Assessment: Thoroughness In Paper Reading:**

I read the paper at least twice and used my best judgement in assessing the paper.

---

> ### Author Response · Authors · 2019-11-15
> **Thank you for your feedback!**
>
> Thank you for the generous acknowledgement of our work! We agree that it is a good idea to add “natural” into the title. However, we are only focusing on Natural Language Understanding tasks in this paper and have not tried deploying such a technology to pretraining language models for better feature representations. Therefore, we have changed the title to “FreeLB: Enhanced Adversarial Training for Natural Language Understanding”.

---

### Official Review · AnonReviewer3 · 2019-11-02
**Official Blind Review #3**

**Rating:** 8

**Review:**

In this paper, the authors present a new adversarial training algorithm and apply it to the fintuning stage large scale language models BERT and RoBERTa. They find that with FreeLB applied to finetuning, both BERT and RoBERTa see small boosts in performance on GLUE, ARC, and CommonsenseQA. The gains they see on GLUE are quite small (0.3 on the GLUE test score for RoBERTa) but the gains are more substantial on ARC and CommonsenseQA. The paper also presents some ablation studies on the use of the same dropout mask across each ascent step of FreeLB, empirically seeing gains by using the same mask. They also present some analysis on robustness in the embedding space, showing that FreeLB leads to greater robustness than other adversarial training methods

This paper is clearly presented and the algorithm shows gains over other methods. I would recommend that the authors try testing their method on SuperGLUE because it's possible they're hitting ceiling issues with GLUE, suppressing any gains the algorithm may yield.

Questions,
-  In tables 4 and 5, why are only results on RTE, CoLA, and MRPC presented? If this is because there was not noticeable difference on the other GLUE datasets, please mention it in the text.
- I realize that this method is meant to increase robustness in the embedding space, but did you do any error analysis on the models? Did they make different types of errors than models fine-tuned the vanilla way?

Couple typos,
- Section 2.2, line 1: many -> much
- Section 4.2, GLUE paragraph: 88 -> 88.8

**Experience Assessment:**

I do not know much about this area.

**Review Assessment: Checking Correctness Of Derivations And Theory:**

I assessed the sensibility of the derivations and theory.

**Review Assessment: Checking Correctness Of Experiments:**

I assessed the sensibility of the experiments.

**Review Assessment: Thoroughness In Paper Reading:**

I read the paper at least twice and used my best judgement in assessing the paper.

---

> ### Author Response · Authors · 2019-11-15
> **Thank you for your feedback!**
>
> Thank you for the acknowledgement of our work and the valuable suggestions! We try to address all of your specific concerns and comments below.
>
> > "- In tables 4 and 5, why are only results on RTE, CoLA, and MRPC presented? If this is because there was not noticeable difference on the other GLUE datasets, please mention it in the text."
> We were not able to finish the experiments on all tasks by the time of submission, since each evaluation is relatively expensive, taking at least 5 runs. During the discussion period, we have focused on providing more results of YOPO on other GLUE tasks, as shown in Table 8 in the Appendix. YOPO is an important variant of adversarial training method to compare with, and the results indicate a deteriorating performance with the increase of shallow-layer updates, something YOPO advocates. We have not been able to finish the experiments for evaluating the effect of variational dropout and comparing the embedding space invariance (Table 5) on the remaining GLUE tasks, but will definitely add them into our next version. We are not able to provide results on SuperGLUE for the moment due to its huge scale, but we have achieved more improvements on CommonsenseQA and ARC. The results have been integrated into the new version.
>
> > "- … did you do any error analysis on the models? Did they make different types of errors than models fine-tuned the vanilla way?"
> We have compared the models with and without FreeLB based on the diagnostic information provided by the GLUE benchmark. For the ensembled RoBERTa-large models, except for Named Entities (35.0/45.9), Quantifiers (63.9/66.1), Common Sense (59.5/60.5), Interval/Numbers (31.3/38.9), Universal (Logic) (75.3/85.0), Relative Clauses (32.8/37.3), FreeLB demonstrates improvements on all the remaining 30 diagnostic metrics (Matthew’s Corr), with the most significant improvements in Morphological Negation (80.8/72.9), Negation (38.8/35.5), Conjunction (74.1/67.3), Disjunction (8.8/-3.1), Existential (Logic) (48.7/42.2), Temporal (Logic) (49.1/41.0), Anaphora/Coreference (54.2/48.8), Coordination Scopes (48.8/41.7).
> For the single BERT-base model, comparing with Jacob Devlin’s submission, except for Lexical Entailment (31.4/35.9), Symmetry/Collectivity (0/26.5), Redundancy (59.2/67.7), Structure Ellipsis/Implicits (35.2/39.4), Structure Datives (53.1/67.3),  Structure Intersectivity (27/30.4), Structure Restrictivity (-19/-13.5), Negation (15.6/24), Conjunction (-12.1/-8.3), Existential (Logic) (32.4/33.7), Downward Monotone (Logic) (-72.9/-66.5), FreeLB demonstrates improvement on all remaining 25 diagnostic metrics. Consistent with results for RoBERTa, in this case FreeLB also shows significant improvements in Anaphora/Coreference (37.2/32.2), Core Args (37.2/29, 51.6/48.9 for RoBERTa), Morphological Negation (64.2/45), and Coordination Scopes (40.9/29.8). We will add this analysis in the final version.

---

### Author Response · Authors · 2019-11-15
**Thank the reviewers for their time!**

We would like to thank the reviewers for their time and their acknowledgement of our work. We have updated a new version, which corrected the spotted typos and includes more experimental results. Generally, by further exploring the hyperparameters, we have improved our highest single-model dev set accuracy from 78.64 to 78.81 on CommonsenseQA, and our ensembled model obtains an accuracy of 73.1, compared with RoBERTa (ensemble model) 72.5 in the leaderboard. We have also improved our dev/test set accuracy on ARC-Easy and ARC-Challenge from 84.56/85.35 (ensemble) and 67.56/67.32 (ensemble) to 84.91/85.44 (single-model) and 70.23/67.75 (single-model), remaining the first place on both leaderboards. We will add more ablation studies on all tasks of GLUE, and test our method on other important benchmarks in our future version.

---

### Decision · Program_Chairs · 2019-12-19

**Decision:**

Accept (Spotlight)

**Comment:**

The paper proposes a new algorithm for adversarial training of language models.  This is an important research area and the paper is well presented, has great empirical results and a novel idea.